# Psychosocial health of school-going adolescents during the COVID-19 pandemic: Findings from a nationwide survey in Bangladesh

**Kamrun Nahar Koly**[1]*, **Md. Saiful Islam**[1], **Marc N. Potenza**[2,3,4,5], **Rashidul Alam Mahumud**[6,7], **Md. Shefatul Islam**[8], **Md. Salim Uddin**[8], **Md. Afzal Hossain Sarwar**[8], **Farzana Begum**[1], **Daniel D. Reidpath**[1,9]

1 Health System & Population Studies Division, International Centre for Diarrhoeal Disease Research, Bangladesh (icddr,b), Mohakhali, Dhaka, Bangladesh, 2 Department of Psychiatry and Child Study Center, Yale School of Medicine, New Haven, CT, United States of America, 3 Connecticut Mental Health Center, New Haven, CT, United States of America, 4 Connecticut Council on Problem Gambling, Wethersfield, CT, United States of America, 5 Department of Neuroscience and Wu Tsai Institute, Yale University, New Haven, CT, United States of America, 6 Faculty of Medicine and Health, NHMRC Clinical Trials Centre, The University of Sydney, Camperdown, NSW, Australia, 7 Centre for Health Research, University of Southern Queensland, Toowoomba, QLD, Australia, 8 Aspire to Innovate (a2i), Information and Communication Technology Division, Agargaon, Dhaka, Bangladesh, 9 Institute for Global Health and Development, Queen Margaret University, Edinburgh, Scotland

* koly@icddrb.org

**Data Availability Statement:** All relevant data are within the paper and its Supporting Information files.

## Abstract

### Background

Common psychosocial health problems (PHPs) have become more prevalent among adolescents globally during the COVID-19 pandemic. However, the psychosocial health of school-going adolescents has remained unexplored in Bangladesh due to limited research during the pandemic. The present study aimed to estimate the prevalence of PHPs (i.e., depression and anxiety) and assess associated lifestyle and behavioral factors among school-going adolescents in Bangladesh during the COVID-19 pandemic.

### Methods

A nationwide cross-sectional survey was conducted among 3,571 school-going adolescents (male: 57.4%, mean age: 14.9±1.8 years; age range: 10–19 years) covering all divisions, including 63 districts in Bangladesh. A semi-structured e-questionnaire, including informed consent and questions related to socio-demographics, lifestyle, academics, pandemic and PHPs, was used to collect data between May and July 2021.

### Results

The prevalence of moderate to severe depression and anxiety were 37.3% and 21.7%, respectively, ranging from 24.7% in the Sylhet Division to 47.5% in the Rajshahi Division for depression, and from 13.4% in the Sylhet Division to 30.3% in the Rajshahi Division for

**Funding:** The author(s) received no specific funding for this work.

**Competing interests:** The authors have declared that no competing interests exist.

anxiety. Depression and anxiety were associated with older age, reports of poor teacher cooperation in online classes, worries due to academic delays, parental comparison of academic performance with other classmates, difficulties coping with quarantine situations, changes in eating habits, weight gain, physical inactivity and having experienced cyberbullying. Moreover, being female was associated with higher odds of depression.

## Conclusions

Adolescent psychosocial problems represent a public health problem. The findings suggest a need for generating improved empirically supported school-based psychosocial support programs involving parents and teachers to ensure the well-being of adolescents in Bangladesh. School-based prevention of psychosocial problems that promote environmental and policy changes related to lifestyle practices and active living should be developed, tested, and implemented.

## Background

Adolescence is an important developmental transition period between childhood and adulthood that includes multiple physical, cognitive and psychosocial changes [1]. In some cases, these changes have lasting direct and indirect negative effects on mental and physical health, academic performance, and subsequent life opportunities [2–4]. Depression and anxiety are the two most common mental health problems during childhood and adolescence [5, 6]. Moreover, the COVID-19 pandemic, and the disease control measures that were implemented created additional, unforeseen stressors for adolescents. Stressors included academic delays, the uncertainty of the future, financial concerns, and worry about becoming infected. These were all reported as additional contributors to mental health issues in adolescents.

Based on pre-COVID data, the World Health Organization (WHO) estimated that mental health problems accounted for 16% of the global burden of disease and injury among people aged 10–19 years [1]. A systematic review and meta-analysis reported that the global pooled prevalence of mental health problems was 13.4% among children and adolescents [7]. While mental health problems in adolescence frequently resolve [8, 9], they can lead to complex and severe mental illness in later life if unsolved [10]. In the transitional period of adolescence, commonly reported mental health problems include depression, anxiety, and attention deficit hyperactivity disorder (ADHD), as well as conduct, eating, psychotic, and substance use disorders [1]. The National Mental Health Survey of Bangladesh 2019 reported a 13.6% overall prevalence of *any* mental health disorders in individuals aged 7–17 years and a 16.8% prevalence in those aged 18 years or older [11]. Moreover, several other pre-COVID (Coronavirus disease 2019) surveys conducted among school-going adolescents in Bangladesh reported frequent anxiety (18.1%), depression (25%-49%), and conduct disorders (8.9%), as well as suicidal behavior (11.7%) [4, 12–16].

COVID-19 was recognized as a global pandemic on March 11, 2020 [17]. To reduce the transmission of the virus, countries around the world implemented various public health measures. In Bangladesh, the measures included social distancing, physical lockdown of areas, and the closure of all educational institutions from March 2020 to September 2021 [18–20]. As a result of the measures, about 38 million students and one million teachers from primary, secondary, and tertiary education institutes were in lockdown [21]. The lockdown period resulted

in decreased physical activity, more screen time, irregular sleep patterns, increased loneliness and poorer dietary habits [22, 23].

Globally, during the pandemic period, increases were reported in multiple mental health conditions among adolescents, including anxiety, depression, trauma, grief, suicidal behaviors, and substance [24, 25]. In Bangladesh, several studies investigating COVID-19 impacts on mental health were conducted [26–37]. However, very few studies looked at mental health in adolescents and instead focused on specific groups (university students, healthcare workers) or populations (slum dwellers, general population). The lack of information about pandemic related mental health problems in adolescents is a critical gap. Understanding the gap can inform the development of interventions aimed at promoting wellbeing during and beyond pandemics. This was, in fact, the motivation for the current study, which was to be used to inform a teacher led intervention supporting adolescent mental health [38, 39].

Given the paucity of data on adolescent mental health in Bangladesh during the pandemic, we sought to estimate the prevalence of the two most common issues (depression and anxiety) as well as their sociodemographic and behavioral correlates. We hypothesized that the prevalence estimates of depression and anxiety would be elevated among school-going adolescents during the COVID-19 pandemic in Bangladesh. We also hypothesized that depression and anxiety would be associated with select socio-demographic, educational, and pandemic related factors.

## Methods

### Study design

A cross-sectional survey was conducted by "Aspire to Innovate" (a2i; https://a2i.gov.bd/), an initiative of the Bangladeshi government. The survey investigated the psychosocial health of the Bangladeshi school-going adolescents during the COVID pandemic [40]. The survey was conducted using a2i's online "edutainment" platform designed to nurture the educational, psychosocial and life skills of school-going adolescents in Bangladesh [41].

### Study setting and participants

a2i is a governmental program in the Information and Communication Technology (ICT) Division of Bangladesh that is supported by the Cabinet Division and United Nations Development Programme (UNDP)[41]. a2i is a program of the government's Digital Bangladesh agenda which has an edutainment online platform named "Kishore Batayan–Konnect" that has been developed for school-going adolescents [41]. a2i aims to nurture the educational, psychosocial, and life skills of school-going adolescents in Bangladesh[42]. Adolescents can share and learn essential life lessons from different creative multimedia content that can help advance their social and personal skills.

Working with researchers from icddr,b (International Centre for Diarrhoeal Disease Research, Bangladesh), a2i developed a survey to identify common psychosocial problems among students and their correlates. The data were used to inform the psychosocial skills training for teachers supporting adolescent mental health.

The inclusion criteria for gparticipants included being: (i) adolescents enrolled in secondary or higher secondary school and enrolled in Konnect, (ii) students of the selected teachers involved in the a2i needs assessment, (iii) willing to participate in the survey with parental consent. The cross-sectional survey was conducted between January and August of 2021. Initially, 3,729 surveys were submitted using the Google Forms. After removing incomplete and missing data, overall 3,571 school-going adolescents from the 63 districts (out of 64, one district

did not have internet connection) of all eight divisions of Bangladesh were included in the final analysis.

## Study variables

A semi-structured questionnaire was developed that included informed consent as well as questions related to socio-demographics, lifestyle, and academic and other factors. Two scales were included to assess depression and anxiety—the Patient Health Questionnaire (PHQ-9) and Generalized Anxiety Disorder (GAD-7). A shareable link using Google Forms was generated for the online survey. The survey was made available to the selected teachers to obtain the data from the selected students who had parents' consent. The survey took approximately 15–20 minutes for each participant.

**Outcome variables.** The two outcome variables were depression measured using the PHQ-9 and anxiety measured using the GAD-7. Both of these scales had been used in previous studies in Bangladesh, including school-going adolescents, and had validated and translated Bangla versions [4, 12, 42, 43].

**Patient Health Questionnaire (PHQ-9).** Participants' levels of depression were assessed using the validated Bangla version of the PHQ-9 scale [44, 45]. The PHQ-9 consists of nine questions assessed using a four-point Likert-type scale (from "0 = not at all" to "3 = almost every day") based on self-reported experiences during the prior two weeks. Total scores are obtained by summing the raw scores of the nine questions, generating a range from 0 to 27 [46]. Higher scores reflect more severe depression. Five predefined cutoff points were used to determine severity levels of depression: i) minimal: 0–4; ii) mild: 5–9; iii) moderate: 10–14; iv) severe: 15–19; and, v) extremely severe: 20 or higher [4, 12, 42, 43, 46]. In analyses, a score of 10 or higher was treated as a positive indicator of moderate to severe depression [4, 12, 42, 43]. The Cronbach alpha of the PHQ-9 was 0.89 in the present study.

**Generalized Anxiety Disorder (GAD-7).** Participants' anxiety levels were assessed using the Bangla version of the GAD-7 scale [47]. The GAD-7 consists of seven questions concerning problems related to symptoms of anxiety over the last two weeks. Responses were recorded on a four-point Likert-type scale (from "0 = not at all" to "3 = almost every day"). Total scores were obtained by summing the raw scores of the seven questions giving an individual score ranging from 0 to 21 [48]. Higher scores reflect more severe anxiety. Four predefined cut-off points were used to determine severity levels of anxiety: i) minimal: 0–4; ii) mild: 5–9; and iii) moderate: 10–14; and iv) severe: 15–21 [12, 42, 43, 47]. During analyses, a score of 10 or higher was treated as a positive indicator of moderate to severe anxiety [12, 42, 43, 47]. The Cronbach alpha of the GAD-7 was 0.84 in the present study.

## Explanatory variables

**Socio-demographic and academic information.** Socio-demographics including age, sex (male/ female), and geographic division (Dhaka/ Barisal/ Chittagong/ Khulna/ Mymensingh/ Rajshahi/ Rangpur/ Sylhet) were obtained from participants. In addition, academic information including school type (secondary school/ higher secondary school) and attending online class (yes/ no) were also recorded. The students were also asked about the cooperation by the teachers in online classes (i.e., never/ sometimes/ always) and also if their parents compare them with their classmates (i.e., none/ some/ much).

**Lifestyle measures.** Participants were asked about their eating behaviors (*"Have your eating behaviors changed during the lockdown period?"*), body weight changes (*"Was there any change in your weight during the lockdown period?"*), and physical exercise (*"Did you engage in any light physical activity during the lockdown period?"*) during the COVID-19-related

lockdown period. This section also included questions about the availability of internet connection at home, ways of communicating with peers during the lockdown, and possible problems during the pandemic (loneliness/ missing friends/ disruption of studies/ poor academic performance).

**Cyberbullying.**   To assess cyberbullying, the participants were asked a single-item question (i.e., *Do your classmates bully you online*?) with three possible responses (i.e., yes/ sometimes/ no).

## Statistical analysis

The dataset was cleaned, coded, and sorted using the Microsoft Excel (version 2019) before being imported into the Statistical Package for the Social Sciences (SPSS, version 25.0). Descriptive statistics, including frequencies and percentages, were computed for categorical variables; means and standard deviations were used for continuous variables. Bivariate analyses (i.e., Chi-square tests or t-tests where appropriate) were performed to explore associations between explanatory and outcome variables. The potential multi-collinearity was checked by using tolerance ($> 0.1$) and variance inflation factor (VIF $< 10$) before performing regression analysis. Variables were significant at $p < 0.05$ in the binary logistic regression analyses were subsequently included in multivariable logistic regression models to determine the factors associated with the outcome variables (depression and anxiety). Crude and adjusted odds ratios (CORs and AORs) were reported from regression models with 95% confidence intervals. A $p$-value less than 0.05 was considered statistically significant in this exploratory study.

## Ethics

This survey was conducted by a2i following the study protocol having been reviewed and approved by the Intuitional Review Board (IRB) of icddr,b. The survey was conducted in accordance with guidelines outlined in the Helsinki declaration. Respondents participated in the survey willingly in an informed fashion without compensation. Students' parents were informed about the survey. The study's objectives, risks and benefits of participation, and options to not participate in the survey were presented in the informed consent section. E-written consent were obtained from participants and their parents.

## Results

### General characteristics of the sample

A total of 3,571 school-going adolescents were included in the final analysis. Students from the 6[th] through 12[th] grades and aged 10–19 years (mean age = 14.90 [SD = 1.80]) participated. Respondents were from 63 of the 64 districts that comprised the eight divisions of Bangladesh. Most respondents were male (57.52%), many came from the Sylhet division (24.67%), and most were from secondary schools (grades 6–10; 85.33%) (Table 1).

### Academic factors during the pandemic

More than a quarter of participants reported poor internet connectivity at their homes (26.1%) (Table 1). About 77.3% of adolescents attended online classes. Most participants reported that their teachers were cooperative when they asked questions in online classes (71.7%). Most reported worries due to academic delays (60.2%), and many reported that their parents compared their academic performances with those of other classmates (44.2%).

**Table 1. Characteristics of participants and distribution of mental health conditions with respect to dependent variables.**

| Variables | Number of participants n (%) | Percentages with depression (n = 1332; 37.30%) | | Percentages with anxiety (n = 774; 21.67%) | | Percentages with depression and anxiety (n = 696; 19.49%) | |
|---|---|---|---|---|---|---|---|
| | | % (95% CI) | p-value | % (95% CI) | p-value | % (95% CI) | p-value |
| *Socio-demographic information* | | | | | | | |
| **Age** (mean ± SD) | 14.90±1.80 | 15.44±1.77 | <0.001‡ | 15.56±1.75 | <0.001‡ | 15.60±1.72 | <0.001‡ |
| **Sex** | | | | | | | |
| Male | 2054 (57.52) | 35.74 (33.68–37.83) | 0.024† | 20.59 (18.89–22.38) | 0.068† | 18.26 (16.63–19.97) | 0.030† |
| Female | 1517 (42.48) | 39.42 (36.98–41.90) | | 23.14 (21.07–25.31) | | 21.16 (19.16–23.27) | |
| **Division** | | | | | | | |
| Dhaka | 865 (24.22) | 41.73 (38.48–45.04) | <0.001† | 23.24 (20.51–26.14) | <0.001† | 21.39 (18.75–24.21) | <0.001† |
| Barisal | 472 (13.22) | 36.44 (32.19–40.85) | | 20.76 (17.29–24.59) | | 18.01 (14.74–21.66) | |
| Chittagong | 441 (12.35) | 38.32 (33.87–42.92) | | 24.26 (20.44–28.42) | | 21.32 (17.69–25.32) | |
| Khulna | 258 (7.22) | 46.12 (40.11–52.22) | | 28.29 (23.06–34.01) | | 26.74 (21.62–32.38) | |
| Mymensingh | 50 (1.40) | 34.00 (22.06–47.74) | | 18.00 (9.30–30.28) | | 18.00 (9.30–30.28) | |
| Rajshahi | 333 (9.33) | 47.45 (42.13–52.81) | | 30.33 (25.58–35.42) | | 26.43 (21.91–31.35) | |
| Rangpur | 271 (7.59) | 43.54 (37.73–49.49) | | 24.72 (19.87–30.11) | | 23.99 (19.2–29.33) | |
| Sylhet | 881 (24.67) | 24.74 (21.98–27.67) | | 13.39 (11.27–15.76) | | 11.46 (9.49–13.69) | |
| *Academic information* | | | | | | | |
| **Academic class** | | | | | | | |
| Secondary school | 3047 (85.33) | 33.51 (31.85–35.20) | <0.001† | 18.84 (17.48–20.26) | <0.001† | 16.77 (15.48–18.13) | <0.001† |
| Higher secondary school | 524 (14.67) | 59.35 (55.10–63.50) | | 38.17 (34.08–42.38) | | 35.31 (31.3–39.47) | |
| **Internet connection at home** | | | | | | | |
| Bad | 933 (26.13) | 50.16 (46.96–53.36) | <0.001† | 32.37 (29.42–35.42) | <0.001† | 29.15 (26.31–32.13) | <0.001† |
| Moderate | 1585 (44.39) | 36.28 (33.94–38.67) | | 20.88 (18.94–22.94) | | 18.93 (17.06–20.91) | |
| Good | 1053 (29.49) | 27.45 (24.81–30.2) | | 13.39 (11.43–15.55) | | 11.78 (9.93–13.83) | |
| **Attending classes online** | | | | | | | |
| Yes | 2761 (77.32) | 33.14 (31.40–34.91) | <0.001† | 18.18 (16.78–19.65) | <0.001† | 16.04 (14.71–17.45) | <0.001† |
| No | 810 (22.68) | 51.48 (48.04–54.91) | | 33.58 (30.39–36.89) | | 31.23 (28.11–34.49) | |
| **Teachers' cooperation while asking questions in class** | | | | | | | |
| Never | 356 (9.97) | 60.39 (55.25–65.37) | <0.001† | 42.70 (37.63–47.88) | <0.001† | 40.17 (35.17–45.32) | <0.001† |
| Sometimes | 654 (18.31) | 46.64 (42.83–50.47) | | 29.05 (25.67–32.62) | | 26.61 (23.33–30.09) | |
| Always | 2561 (71.72) | 31.71 (29.93–33.53) | | 16.87 (15.46–18.36) | | 14.80 (13.46–16.21) | |
| **Parental comparison of academic performance with other classmates** | | | | | | | |
| None | 1192 (33.38) | 23.66 (21.31–26.13) | <0.001† | 13.26 (11.42–15.27) | <0.001† | 10.99 (9.31–12.86) | <0.001† |
| Some | 800 (22.40) | 34 (30.78–37.34) | | 19.25 (16.63–22.09) | | 17.00 (14.52–19.72) | |
| Much | 1579 (44.22) | 49.27 (46.81–51.74) | | 29.26 (27.05–31.54) | | 27.17 (25.02–29.4) | |
| **Worries due to academic delay** | | | | | | | |
| None | 448 (12.55) | 16.74 (13.5–20.41) | <0.001† | 10.49 (7.91–13.58) | <0.001† | 7.14 (5.03–9.81) | <0.001† |
| Some | 975 (27.30) | 18.97 (16.61–21.53) | | 9.44 (7.72–11.39) | | 7.59 (6.05–9.38) | |
| Much | 2148 (60.15) | 49.91 (47.79–52.02) | | 29.56 (27.66–31.52) | | 27.47 (25.61–29.38) | |
| *Lifestyle during the pandemic* | | | | | | | |
| **Coping with the quarantine situation** | | | | | | | |
| Poorly | 1225 (34.30) | 54.86 (52.06–57.63) | <0.001† | 36.00 (33.35–38.72) | <0.001† | 33.55 (30.95–36.23) | <0.001† |
| Moderately | 1859 (52.06) | 30.23 (28.18–32.35) | | 14.63 (13.08–16.29) | | 12.69 (11.24–14.27) | |
| Well | 487 (13.64) | 20.12 (16.75–23.85) | | 12.53 (9.81–15.69) | | 10.06 (7.63–12.97) | |
| **Feeling loneliness** | | | | | | | |
| No | 2002 (56.06) | 37.76 (35.66–39.9) | 0.519† | 22.38 (20.59–24.24) | 0.249† | 20.13 (18.42–21.93) | 0.276† |
| Yes | 1569 (43.94) | 36.71 (34.35–39.12) | | 20.78 (18.83–22.84) | | 18.67 (16.8–20.66) | |

*(Continued)*

**Table 1.** (Continued)

| Variables | Number of participants n (%) | Percentages with depression (n = 1332; 37.30%) | | Percentages with anxiety (n = 774; 21.67%) | | Percentages with depression and anxiety (n = 696; 19.49%) | |
|---|---|---|---|---|---|---|---|
| | | % (95% CI) | *p*-value | % (95% CI) | *p*-value | % (95% CI) | *p*-value |
| **Eating habits changed during the locked down period** | | | | | | | |
| Yes | 1378 (38.59) | 46.81 (44.18–49.45) | <0.001[†] | 30.70 (28.3–33.17) | <0.001[†] | 27.72 (25.41–30.13) | <0.001[†] |
| Somewhat | 1134 (31.76) | 36.60 (33.83–39.43) | | 19.40 (17.18–21.78) | | 17.81 (15.67–20.12) | |
| No | 1059 (29.66) | 25.68 (23.12–28.38) | | 12.37 (10.49–14.46) | | 10.58 (8.83–12.54) | |
| **Weight changes during pandemic (self-reported)** | | | | | | | |
| Loss | 896 (25.09) | 42.41 (39.2–45.67) | <0.001[†] | 23.33 (20.65–26.18) | <0.001[†] | 20.76 (18.2–23.51) | <0.001[†] |
| Gain | 623 (17.45) | 55.86 (51.94–59.72) | | 37.88 (34.14–41.74) | | 34.83 (31.17–38.64) | |
| Unaware | 1397 (39.12) | 31.07 (28.68–33.53) | | 17.75 (15.82–19.82) | | 16.11 (14.25–18.1) | |
| No change | 655 (18.34) | 25.95 (22.71–29.41) | | 12.37 (10.01–15.05) | | 10.38 (8.22–12.89) | |
| **Physical activity** | | | | | | | |
| Yes | 2100 (58.81) | 30.57 (28.63–32.57) | <0.001[†] | 16.24 (14.71–17.86) | <0.001[†] | 14.52 (13.07–16.08) | <0.001[†] |
| No | 1471 (41.19) | 46.91 (44.36–49.46) | | 29.44 (27.15–31.80) | | 26.58 (24.37–28.88) | |
| ***Victimization*** | | | | | | | |
| **Cyberbullying** | | | | | | | |
| Yes | 567 (15.88) | 63.32 (59.29–67.21) | <0.001[†] | 41.98 (37.96–46.07) | <0.001[†] | 39.86 (35.89–43.93) | <0.001[†] |
| No | 3004 (84.12) | 32.39 (30.73–34.08) | | 17.84 (16.51–19.24) | | 15.65 (14.38–16.98) | |

Note:

[†]Chi-square test

[‡]t-test.

## Lifestyle during the pandemic

Many adolescents (34.3%) reported not coping well with lockdown during the pandemic (Table 1). Many reported feeling lonely (43.9%), not participating in physical exercise (41.2%), and changing their eating behaviors (38.6%) during the lockdown period. Many participants reported weight loss (25.1%) or weight gain (17.5%), and others reported no change (18.3%) or being unaware of changes in body weight (39.1%). A number of participants experienced cyberbullying from their classmates (15.8%).

## Prevalence of depression and anxiety

Of the participants, 37.3% reported moderate to extremely severe symptoms of depression; 62.7% reported estimates of no or mild symptoms of depression (Fig 1). Of the participants, 21.7% reported moderate to severe symptoms of anxiety; 78.3% reported estimates of no or mild symptoms of anxiety (Fig 1).

Nearly one-fifth (19.5%) of participants experienced both depression and anxiety, whereas 17.8% had only depression and 2.2% had only anxiety (Fig 2).

The conditional distribution of symptoms of depression and anxiety by age is shown in Fig 3.

## Factors associated with depression and anxiety

In bivariate analyses, depression and anxiety were each associated at $p < 0.05$ with most variables (Table 1). Multivariate logistic regression analyses were performed including all significant variables ($p < 0.05$) from binary logistic regression analyses (Table 2). The values of Cox & Snell R Square, and Nagelkerke R Square for multivariable logistic regression of depression were 0.27, and 0.36, respectively. Depression was associated with age (AOR: 1.12; 95% CI: 1.05–1.19,

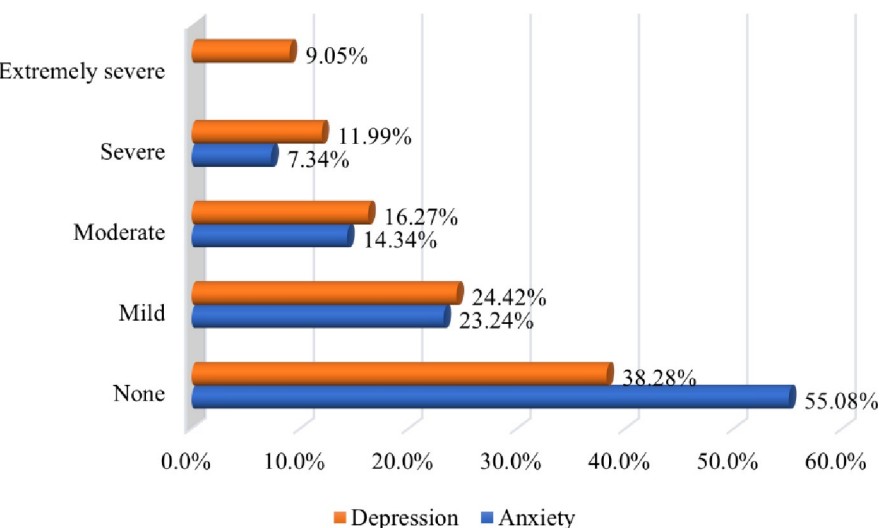

**Fig 1. Participants' anxiety and depression levels.**

$p < 0.001$), female gender (AOR: 1.26, 95% CI: 1.05–1.50, $p = 0.011$), higher secondary school status (AOR: 1.50; 95% CI: 1.14–1.98, $p = 0.004$), and reports of teachers not being cooperative when students asked questions in online classes (AOR: 1.74; 95% CI: 1.32–2.29, $p < 0.001$), poor internet connection at home (AOR = 1.46; 95% CI: 1.12–1.89, $p = 0.005$), worries about academic delay (AOR: 3.55; 95% CI: 2.62–4.82, $p < 0.001$), parental comparisons of their academic performances with those of other classmates (AOR: 2.15; 95% CI: 1.77–2.61, $p < 0.001$), difficulties coping with the quarantine situation (AOR: 2.96; 95% CI: 2.21–3.95, $p < 0.001$), changes in eating behaviors during the pandemic (AOR: 1.55; 95% CI: 1.26–1.91, $p < 0.001$), weight gain (AOR: 2.09; 95% CI: 1.59–2.75, $p < 0.001$), physical inactivity (AOR: 1.51; 95% CI: 1.28–1.78, $p < 0.001$) and experiencing cyberbullying (AOR: 2.67; 95% CI: 2.16–3.35, $p < 0.001$).

The values of Cox & Snell R Square, and Nagelkerke R Square for multivariable logistic regression of anxiety were 0.20, and 0.31, respectively. Anxiety was associated with age (AOR:

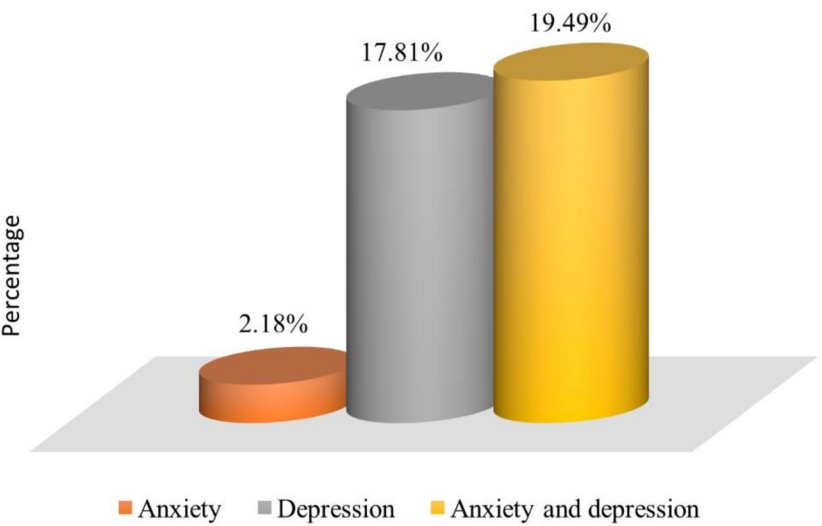

**Fig 2. Presence of psychosocial health problems.**

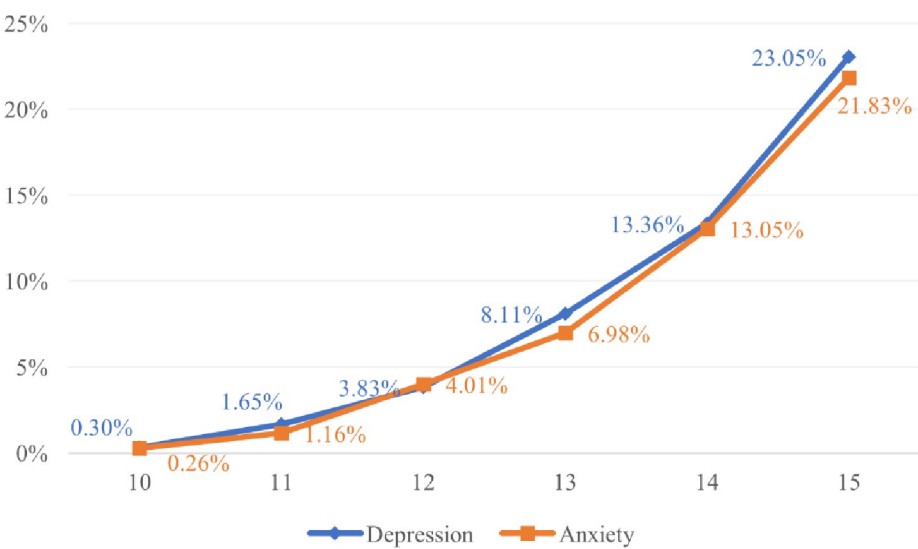

**Fig 3. Distribution of depression and anxiety with participants' age (from 10 to 15 years).**

1.10; 95% CI: 1.03–1.18, $p = 0.007$) and inversely linked to living in the Sylhet division as compared to the Dhaka division (AOR: 0.66; 95% CI: 0.51–0.84, $p = 0.001$). Anxiety was associated with higher secondary school status (AOR: 1.39; 95% CI: 1.03–1.86, $p = 0.029$) and reports of teachers not being cooperative when students asked questions in online classes (AOR: 1.91; 95% CI: 1.45–2.52, $p < 0.001$), worries about academic delays (AOR: 2.50; 95% CI: 1.75–3.58, $p < 0.001$), parental comparisons of their academic performances with those of other classmates (AOR: 1.69; 95% CI: 1.34–2.12, $p < 0.001$), difficulties coping with the quarantine situation (AOR: 2.16; 95% CI: 1.55–3.01, $p < 0.001$), changes in eating behaviors during the pandemic (AOR: 2.01; 95% CI: 1.57–2.56, $p < 0.001$), weight gain (AOR: 2.32; 95% CI: 1.68–3.19, $p < 0.001$), physical inactivity (AOR: 1.60; 95% CI: 1.33–1.92, $p < 0.001$), and experiencing cyberbullying (AOR: 2.24; 95% CI: 1.80–2.80, $p < 0.001$).

The values of Cox & Snell R Square, and Nagelkerke R Square for multivariable logistic regression models of depression and anxiety were 0.21, and 0.33, respectively. Adolescents with both depression and anxiety were more likely to be older, female, and living in the Dhaka division. They were also more likely to report having a poor internet connection at home, lack of teacher cooperation when asking questions in online classes, worries about academic delays, difficulties coping with the quarantine situation, parental comparisons of their academic performances with those of other classmates, changes in eating behaviors, physical inactivity, and cyberbullying.

## Discussion

Depression and anxiety are often experienced by adolescents and may compromise their overall wellbeing and academic functioning [49, 50]. Moreover, the COVID-19 pandemic has further increased the mental health concerns. The present study represents the largest nationwide survey by the Bangladesh government examining psychosocial health correlates of school-going adolescents during the pandemic. Many adolescents experienced moderate to severe depression (37.3%) and anxiety (21.7%). Participants who reported being older, being from higher secondary classes, residing in the Dhaka division or area, perceiving a lack of teacher cooperation in class, having worries about academic delay, having perceived parental

**Table 2. Regression analysis by mental health conditions.**

| Variables | Depression | | | | Anxiety | | | | Depression and anxiety | | | |
|---|---|---|---|---|---|---|---|---|---|---|---|---|
| | Unadjusted | | Adjusted | | Unadjusted | | Adjusted | | Unadjusted | | Adjusted | |
| | OR (95% CI) | p-value | OR (95% CI) | p-value | OR (95% CI) | p-value | OR (95% CI) | p-value | OR (95% CI) | p-value | OR (95% CI) | p-value |
| **Socio-demographic information** | | | | | | | | | | | | |
| **Age** | 1.32 (1.27–1.37) | <0.001 | 1.12 (1.06–1.19) | <0.001 | 1.30 (1.25–1.37) | <0.001 | 1.10 (1.03–1.18) | 0.007 | 1.32 (1.25–1.38) | <0.001 | 1.10 (1.02–1.18) | 0.011 |
| **Sex** | | | | | | | | | | | | |
| Male | Ref. | | Ref. | | Ref. | | — | — | Ref. | | Ref. | |
| Female | 1.17 (1.02–1.34) | 0.024 | 1.26 (1.05–1.50) | 0.011 | 1.16 (0.99–1.36) | 0.068 | — | — | 1.20 (1.02–1.42) | 0.031 | 1.26 (1.02–1.55) | 0.029 |
| **Division** | | | | | | | | | | | | |
| Dhaka | Ref. | | Ref. | | Ref. | | Ref. | | Ref. | | Ref. | |
| Barisal | 0.80 (0.64–1.01) | 0.059 | 1.02 (0.77–1.34) | 0.918 | 0.87 (0.66–1.14) | 0.300 | 1.04 (0.76–1.43) | 0.794 | 0.81 (0.61–1.07) | 0.142 | 1.00 (0.71–1.39) | 0.984 |
| Chittagong | 0.87 (0.67–1.10) | 0.235 | 0.73 (0.55–0.97) | 0.031 | 1.06 (0.81–1.39) | 0.680 | 1.01 (0.73–1.37) | 0.976 | 1.00 (0.75–1.32) | 0.976 | 0.90 (0.65–1.25) | 0.514 |
| Khulna | 1.20 (0.90–1.58) | 0.211 | 0.84 (0.60–1.18) | 0.326 | 1.30 (0.95–1.78) | 0.097 | 0.93 (0.64–1.34) | 0.679 | 1.34 (0.97–1.85) | 0.072 | 0.97 (0.66–1.41) | 0.852 |
| Mymensingh | 0.72 (0.40–1.31) | 0.282 | 0.56 (0.28–1.15) | 0.115 | 0.73 (0.35–1.52) | 0.394 | 0.53 (0.23–1.21) | 0.132 | 0.81 (0.39–1.69) | 0.570 | 0.62 (0.27–1.44) | 0.265 |
| Rajshahi | 1.26 (0.98–1.63) | 0.074 | 1.28 (0.95–1.73) | 0.102 | 1.44 (1.09–1.91) | 0.012 | 1.51 (1.09–2.09) | 0.013 | 1.32 (0.99–1.77) | 0.063 | 1.33 (0.94–1.87) | 0.104 |
| Rangpur | 1.08 (0.82–1.42) | 0.599 | 0.62 (0.44–0.86) | 0.005 | 1.09 (0.79–1.49) | 0.615 | 0.63 (0.44–0.93) | 0.018 | 1.16 (0.84–1.60) | 0.368 | 0.70 (0.47–1.02) | 0.062 |
| Sylhet | 0.46 (0.37–0.56) | <0.001 | 0.66 (0.51–0.84) | 0.001 | 0.51 (0.40–0.66) | <0.001 | 0.77 (0.58–1.03) | 0.082 | 0.48 (0.37–0.62) | <0.001 | 0.68 (0.50–0.93) | 0.014 |
| **Academic information** | | | | | | | | | | | | |
| **Academic class** | | | | | | | | | | | | |
| Secondary school | Ref. | | Ref. | | Ref. | | Ref. | | Ref. | | Ref. | |
| Higher secondary school | 2.98 (2.40–3.50) | <0.001 | 1.50 (1.14–1.98) | 0.004 | 2.66 (2.18–3.24) | <0.001 | 1.39 (1.03–1.86) | 0.029 | 2.77 (2.21–3.32) | <0.001 | 1.32 (0.97–1.79) | 0.076 |
| **Internet connection at home** | | | | | | | | | | | | |
| Bad | 2.66 (2.21–3.21) | <0.001 | 1.24 (0.99–1.56) | 0.061 | 3.10 (2.47–3.87) | <0.001 | 1.46 (1.12–1.89) | 0.005 | 3.08 (2.44–3.90) | <0.001 | 1.38 (1.04–1.82) | 0.024 |
| Moderate | 1.51 (1.27–1.78) | <0.001 | 1.03 (0.84–1.26) | 0.768 | 1.71 (1.38–2.12) | <0.001 | 1.18 (0.93–1.51) | 0.175 | 1.75 (1.40–2.19) | <0.001 | 1.19 (0.92–1.53) | 0.197 |
| Good | Ref. | | Ref. | | Ref. | | Ref. | | Ref. | | Ref. | |
| **Attending classes online** | | | | | | | | | | | | |
| Yes | Ref. | | Ref. | | Ref. | | Ref. | | Ref. | | Ref. | |
| No | 2.14 (1.83–2.51) | <0.001 | 1.17 (0.96–1.43) | 0.130 | 2.28 (1.91–2.71) | <0.001 | 1.21 (0.98–1.51) | 0.082 | 2.38 (1.99–2.85) | <0.001 | 1.24 (0.99–1.55) | 0.067 |
| **Teachers' cooperation while asking questions in class** | | | | | | | | | | | | |
| Never | 3.28 (2.61–4.13) | <0.001 | 1.74 (1.32–2.29) | <0.001 | 3.67 (2.91–4.64) | <0.001 | 1.91 (1.45–2.52) | <0.001 | 3.87 (3.05–4.91) | <0.001 | 2.01 (1.51–2.66) | <0.001 |
| Sometimes | 1.88 (1.58–2.24) | <0.001 | 1.32 (1.08–1.63) | 0.008 | 2.02 (1.66–2.46) | <0.001 | 1.46 (1.16–1.84) | 0.001 | 2.09 (1.70–2.56) | <0.001 | 1.46 (1.15–1.85) | 0.002 |
| Always | Ref. | | Ref. | | Ref. | | Ref. | | Ref. | | Ref. | |
| **Parental comparison of academic performance with other classmates** | | | | | | | | | | | | |
| None | Ref. | | Ref. | | Ref. | | Ref. | | Ref. | | Ref. | |
| Some | 1.66 (1.36–2.03) | <0.001 | 1.35 (1.07–1.70) | 0.010 | 1.56 (1.22–1.99) | <0.001 | 1.25 (0.95–1.64) | 0.110 | 1.66 (1.28–2.15) | <0.001 | 1.32 (0.99–1.77) | 0.063 |

*(Continued)*

**Table 2.** (Continued)

| Variables | Depression | | | | | | Anxiety | | | | | | Depression and anxiety | | | | | |
|---|---|---|---|---|---|---|---|---|---|---|---|---|---|---|---|---|---|---|
| | Unadjusted | | | Adjusted | | | Unadjusted | | | Adjusted | | | Unadjusted | | | Adjusted | | |
| | OR (95% CI) | p-value | | OR (95% CI) | p-value | | OR (95% CI) | p-value | | OR (95% CI) | p-value | | OR (95% CI) | p-value | | OR (95% CI) | p-value | |
| Much | 3.13 (2.66–3.70) | <0.001 | | 2.15 (1.77–2.61) | <0.001 | | 2.71 (2.22–3.30) | <0.001 | | 1.69 (1.34–2.12) | <0.001 | | 3.02 (2.44–3.74) | <0.001 | | 1.91 (1.50–2.44) | <0.001 | |
| **Worries due to academic delay** | | | | | | | | | | | | | | | | | | |
| None | Ref. | | | Ref. | | | Ref. | | | Ref. | | | Ref. | | | Ref. | | |
| Some | 1.17 (0.87–1.57) | 0.312 | | 1.23 (0.88–1.72) | 0.225 | | 0.89 (0.61–1.29) | 0.534 | | 1.02 (0.68–1.54) | 0.917 | | 1.07 (0.69–1.64) | 0.766 | | 1.26 (0.79–2.02) | 0.339 | |
| Much | 4.96 (3.81–6.44) | <0.001 | | 3.55 (2.62–4.82) | <0.001 | | 3.58 (2.61–4.91) | <0.001 | | 2.50 (1.75–3.58) | <0.001 | | 4.92 (3.39–7.14) | <0.001 | | 3.46 (2.28–5.25) | <0.001 | |
| *Lifestyle factors* | | | | | | | | | | | | | | | | | | |
| **Coping with the quarantine situation** | | | | | | | | | | | | | | | | | | |
| Poorly | 4.82 (3.76–6.18) | <0.001 | | 2.96 (2.21–3.95) | <0.001 | | 3.93 (2.93–5.26) | <0.001 | | 2.16 (1.55–3.01) | <0.001 | | 4.51 (3.28–6.20) | <0.001 | | 2.36 (1.64–3.38) | <0.001 | |
| Moderately | 1.72 (1.35–2.19) | <0.001 | | 1.38 (1.04–1.82) | 0.024 | | 1.20 (0.89–1.61) | 0.236 | | 0.91 (0.65–1.26) | 0.568 | | 1.30 (0.94–1.80) | 0.114 | | 0.95 (0.66–1.36) | 0.786 | |
| Well | Ref. | | | Ref. | | | Ref. | | | Ref. | | | Ref. | | | Ref. | | |
| **Feeling loneliness** | | | | | | | | | | | | | | | | | | |
| No | Ref. | | | — | — | | Ref. | | | — | — | | Ref. | | | — | — | |
| Yes | 0.96 (0.83–1.10) | 0.519 | | — | — | | 0.91 (0.77–1.07) | 0.250 | | — | — | | 0.91 (0.77–1.08) | 0.276 | | — | — | |
| **Worries due to academics** | | | | | | | | | | | | | | | | | | |
| None | Ref. | | | Ref. | | | Ref. | | | Ref. | | | Ref. | | | Ref. | | |
| Some | 1.17 (0.87–1.57) | 0.312 | | 1.23 (0.88–1.72) | 0.225 | | 0.89 (0.61–1.29) | 0.534 | | 1.02 (0.68–1.54) | 0.917 | | 1.07 (0.69–1.64) | 0.766 | | 1.26 (0.79–2.02) | 0.339 | |
| Much | 4.96 (3.81–6.44) | <0.001 | | 3.55 (2.62–4.82) | <0.001 | | 3.58 (2.61–4.91) | <0.001 | | 2.50 (1.75–3.58) | <0.001 | | 4.92 (3.39–7.14) | <0.001 | | 3.46 (2.28–5.25) | <0.001 | |
| **Eating habits changed during the locked down period** | | | | | | | | | | | | | | | | | | |
| Yes | 2.55 (2.14–3.03) | <0.001 | | 1.55 (1.26–1.91) | <0.001 | | 3.14 (2.53–3.89) | <0.001 | | 2.01 (1.57–2.56) | <0.001 | | 3.24 (2.58–4.08) | <0.001 | | 1.97 (1.52–2.56) | <0.001 | |
| Somewhat | 1.67 (1.39–2.01) | <0.001 | | 1.42 (1.15–1.76) | 0.001 | | 1.71 (1.35–2.16) | <0.001 | | 1.41 (1.09–1.83) | 0.010 | | 1.83 (1.43–2.35) | <0.001 | | 1.48 (1.12–1.96) | 0.006 | |
| No | Ref. | | | Ref. | | | Ref. | | | Ref. | | | Ref. | | | Ref. | | |
| **Weight changes during pandemic (self-reported)** | | | | | | | | | | | | | | | | | | |
| Loss | 2.10 (1.69–2.62) | <0.001 | | 1.49 (1.15–1.92) | 0.003 | | 2.16 (1.63–2.85) | <0.001 | | 1.36 (0.99–1.86) | 0.056 | | 2.26 (1.68–3.05) | <0.001 | | 1.41 (1.00–1.97) | 0.048 | |
| Gain | 3.61 (2.85–4.57) | <0.001 | | 2.09 (1.59–2.75) | <0.001 | | 4.32 (3.26–5.74) | <0.001 | | 2.32 (1.68–3.19) | <0.001 | | 4.61 (3.42–6.23) | <0.001 | | 2.45 (1.74–3.44) | <0.001 | |
| Unaware | 1.29 (1.04–1.58) | 0.018 | | 0.97 (0.76–1.24) | 0.814 | | 1.53 (1.17–2.00) | 0.002 | | 1.13 (0.84–1.53) | 0.410 | | 1.66 (1.24–2.21) | 0.001 | | 1.21 (0.88–1.67) | 0.237 | |
| No change | Ref. | | | Ref. | | | Ref. | | | Ref. | | | Ref. | | | Ref. | | |
| **Physical activity** | | | | | | | | | | | | | | | | | | |
| Yes | Ref. | | | Ref. | | | Ref. | | | Ref. | | | Ref. | | | Ref. | | |
| No | 2.01 (1.75–2.30) | <0.001 | | 1.51 (1.28–1.78) | <0.001 | | 2.15 (1.83–2.53) | <0.001 | | 1.60 (1.33–1.92) | <0.001 | | 2.13 (1.80–2.52) | <0.001 | | 1.50 (1.24–1.83) | <0.001 | |
| *Victimization* | | | | | | | | | | | | | | | | | | |
| **Cyberbullying** | | | | | | | | | | | | | | | | | | |
| Yes | 3.60 (2.99–4.34) | <0.001 | | 2.67 (2.16–3.35) | <0.001 | | 3.33 (2.75–4.03) | <0.001 | | 2.24 (1.80–2.80) | <0.001 | | 3.57 (2.94–4.34) | <0.001 | | 2.52 (2.00–3.17) | <0.001 | |
| No | Ref. | | | Ref. | | | Ref. | | | Ref. | | | Ref. | | | Ref. | | |

comparison of students' academic performances with those of other classmates, having difficulties coping with quarantine situations, changes in eating behaviors, gaining weight, physical inactivity, or having experienced cyberbullying were more likely to experience depression and anxiety. Being female and loosing weight during the quarantine were also associated with depression, whereas having poor internet connections at home was associated with anxiety. Implications are discussed below.

In the present study, and in comparison to pre-COVID findings among Bangladeshi school-going adolescents using a comparable methodology [12], the percentages of students experiencing moderate to severe depression (37.3% vs. 26.5%) and anxiety (21.7% vs. 18.1%) were higher during the pandemic. Other pre-pandemic findings similarly reported slightly lower estimates of depression (25%-36.6%) [4, 13, 51] and other mental health concerns (13.4–22.9%) among children (< 18 years) [52]. A meta-analysis conducted during the pandemic involving twenty-nine studies sampling 80,879 children and adolescents reported a pooled prevalence estimate of depression of 25.2% (95% CI: 21.2%-29.7%) and anxiety of 20.5% (95% CI: 17.2%-24.4%). These estimates were slightly lower than were found in the present study [6]. Other studies conducted among school-going adolescents using the PHQ-9 and GAD-7 also have observed frequent reports of moderate to severe depression and anxiety during the COVID-19 pandemic including studies from China (depression = 17.3%, anxiety = 10.4%) [53], and the United States (depression = 32%, anxiety = 31%) [54], consistent with meta-analytic findings from 29 studies (depression = 25.2%, anxiety = 20.5%) [6]. The high prevalence estimates of depression and anxiety in the current study may reflect experiences during the COVID-19 pandemic, including social isolation, limited peer interactions, online education (e.g., uncertainty, difficulties utilizing online platforms, challenges in complying with online learning standards), reduced contact with buffering supports (e.g., teachers, coaches), and other factors [6, 51, 55]. However, a pre-COVID study conducted in 2012 among 165 adolescents aged 15–19 years selected from two urban schools in Bangladesh reported a higher prevalence of depression (49% vs. 21.7%) compared to the present study [14]. These differences may reflect differences in assessment instruments and sample characteristics [14]. In the current study, nearly one-fifth (19.5%) of participants experienced both depression and anxiety, consistent with the frequent co-occurrence of the two [12, 42, 43].

In the current study, participants' age was positively associated with depression and anxiety. This is consistent with both the pre-COVID and pandemic research from Bangladesh and elsewhere [3, 12, 13, 51]. Common mental health problems (e.g., depression, anxiety, behavioral problems) are more prevalent in older children associated with puberty- and hormone-related physical and psychological changes [1, 56]. Moreover, social isolation and physical distancing might have impacted older children, who may rely more heavily on peer socialization [6, 57, 58]. However, this present finding contrasts with other reports from Bangladesh that did not find associations between age and depression among school-going adolescents [4, 12, 59].

In the present study, girls were more likely than boys to have higher odds of depression, which is consistent with earlier reports from Bangladesh [3, 4]. A previous study in China among school-going adolescents also reported finding that females had higher prevalence of depression and anxiety compared to males [60]. The finding is also in line with other reports [61–63]. Moreover, cultural practices and gender norms (such as adolescents not being permitted to discuss their pubertal changes with their parents, needing to contribute in household activities, and possibilities of early marriage, violence, and sexual harassment) in South-Asian societies may make adolescent girls more vulnerable to mental health conditions [64].

Participants from the Sylhet division (northeastern Bangladesh) were less likely to have depression and anxiety than those from the Dhaka division in the present study. This finding resonates with previous Bangladeshi reports of participants residing outside the Dhaka

division having had lower likelihoods of depression [65]. One possible reason would be a higher percentage of total COVID-19 cases and deaths in the Dhaka division compared to other divisions, with cases and deaths having been frequently broadcast in electronic and print media [66]. Such information may have increased anxieties about oneself or family members becoming infected and about restrictions on usual movement, travel, and other social activities [20]. Following this rationale, public health awareness messages should be disseminated carefully to reduce unnecessary anxiety. In addition, school-based psychosocial support programs should consider vulnerabilities relating to older age, being female, and geographical location.

Higher secondary school students were more likely to have depression than those secondary school students in the present study, which corroborates previous findings [4]. One possible explanation could be that higher secondary school-going adolescents may have more academic pressure related to getting admission offers into universities based on their performance on the national board exam (Higher Secondary School Certificate [HSC]) in the 12th class in Bangladesh [67]. This additional pressure may predispose them to an increased risk of depression and anxiety [51]. Moreover, Higher secondary school students are typically older compared to other age groups which were taken under consideration. They usually experience more physical, mental and social changes and sometimes burdened with familial responsibilities. These additional stressors might increase the risk of developing depression and anxiety among this group [6, 57, 58].

This study also found a higher odds of depression and anxiety among students who reported worries due to academic delays. A previous study conducted among adolescents and young adults during the COVID-19 pandemic reported that increased anxiety and depression symptoms were linked to increased fear of getting COVID-19 and school-related problems [68]. Furthermore, individuals who reported difficulties coping with lockdown during the pandemic were more likely to experience depression and anxiety. The findings corroborate with the previous reports suggesting that pandemic-related issues such as quarantines, physical distancing, isolation, and educational and economic factors may predispose individuals to common mental health problems including depression and anxiety [20, 28–31, 69]. Moreover, all students experienced academic disruptions (e.g., campus closure, exam postponement) related to the sudden shutdown of all Bangladeshi educational institutions due to the COVID-19 pandemic that began on March 17, 2020 [70]. Further research is needed to investigate how best to address and mitigate academic delays and concerns related to the pandemic, mental health concerns and improve mechanisms for coping with quarantines during pandemics.

In the present study, participants who reported perceived parental comparison of academic performance with other classmates and not having teachers' cooperation while asking questions in class were significantly more likely to experience depression and anxiety. Some previous studies also showed that students reporting more parental pressure to study experienced more depression, anxiety, and stress [71, 72]. Teacher support has been linked to better student mental health and higher resilience [73]. These findings suggest that teacher-mediated school-based support programs may help improve adolescent mental health.

In the present study, participants who did not engage in physical exercise were more likely to have both depression and anxiety. This finding resonates with pre-COVID Bangladeshi data collected from adolescents [4], and another study conducted in China during the COVID-19 pandemic [74]. Physical activity may help adolescents with self-regulation and coping, facilitating mental optimism [74, 75].

Changed eating habits and increased body weight during the pandemic were associated with depression and anxiety, consistent with a prior study conducted among Egyptian youths during the COVID-19, which linked dietary changes to depression and anxiety [76]. Further, a previous study conducted among school-going adolescents in the Indian Kashmir valley

during the pandemic reported that being overweight was associated with depression and anxiety [76]. Similarly, a previous Bangladeshi study including a longitudinal study also found that increased body weight or weight gain was linked to depression and anxiety [77, 78]. Thus, programs promoting physical exercise and maintenance of healthy eating habits and limiting weight gain during pandemics should be developed and tested.

Of the participants, 15.8% reported experiencing cyberbullying from their classmates. Participants who experienced cyberbullying had approximately twice the odds of depression and anxiety than those who did not. This finding is consistent with several recent reports indicating that adolescents who experienced cyberbullying were more likely to experience depression, anxiety, loneliness, suicidal behavior, and psychosomatic symptoms [79, 80]. A review article found a considerable increase of cyberbullying among children and adolescents during the COVID-19 pandemic, with symptoms of anxiety, depression, and suicide ideation [81]. Thus, cyberbullying should be prevented through various mechanisms including parental and school-based education, and appropriate legislation [82].

## Strengths and limitations

The present study was the first nationwide, Bangladeshi-government-led epidemiological survey of psychosocial health among school-going adolescents during the COVID-19 pandemic. More than 3,500 adolescents from all divisions of Bangladesh, including 63 districts, participated.

However, the study has some limitations. First, as the study was cross-sectional in nature, causal relations cannot be established. Future longitudinal studies are warranted. Second, the study used self-reported measures that are vulnerable to multiple biases, including ones related to recall and social desirability. Although over 3,500 school-going adolescents participated, the study may not be regarded as a representative because it utilized an online survey.

## Conclusions and recommendations

The present study revealed high prevalence estimates of depression and anxiety among school-going adolescents during the COVID-19 pandemic in Bangladesh. It also highlighted multiple significant relationships between depression and anxiety and socio-demographic, lifestyle, academic and pandemic measures. The findings raise the possibility that initiating psychosocial support programs among school-going adolescents during pandemics may help protect them from common mental health conditions. Adolescents' perceptions regarding parental and teacher support appear important when considering adolescent depression and anxiety; thus, positive parent-child and teacher-student interactions from adolescent perspectives are important to understand better. Virtual awareness programs could be considered in order to promote physical exercise, healthy eating habits and avoid weight gain. Addressing cyberbullying also is important, and this may best be achieved through collaboration among multiple stakeholders (e.g., parents, teachers, students, governmental groups, healthcare providers). Last but not least, the findings may contribute as baseline information for future research, including longitudinal or interventional studies.

## Supporting information

**S1 File. Dataset.**
(XLSX)

## Acknowledgments

The authors would like to express the most profound gratitude to the authorities, teachers, parents, and students under the *Konnect* platform of a2i who contributed in the study. icddr,b is grateful to the governments of Bangladesh, Canada, Sweden and the UK for providing unrestricted support.

## Author Contributions

**Conceptualization:** Kamrun Nahar Koly.

**Data curation:** Kamrun Nahar Koly, Md. Saiful Islam, Md. Shefatul Islam, Md. Salim Uddin, Md. Afzal Hossain Sarwar.

**Formal analysis:** Kamrun Nahar Koly, Md. Saiful Islam.

**Investigation:** Kamrun Nahar Koly.

**Methodology:** Kamrun Nahar Koly, Md. Saiful Islam.

**Supervision:** Kamrun Nahar Koly.

**Validation:** Kamrun Nahar Koly, Md. Saiful Islam, Marc N. Potenza, Rashidul Alam Mahumud, Md. Shefatul Islam, Md. Salim Uddin, Farzana Begum, Daniel D. Reidpath.

**Writing – original draft:** Kamrun Nahar Koly, Md. Saiful Islam.

**Writing – review & editing:** Kamrun Nahar Koly, Marc N. Potenza, Rashidul Alam Mahumud, Md. Shefatul Islam, Md. Salim Uddin, Md. Afzal Hossain Sarwar, Farzana Begum, Daniel D. Reidpath.

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
