## [Decision Letter · Decision Letter 0]

23 Nov 2022

PONE-D-22-07306Psychosocial health of school-going adolescents during the COVID-19 pandemic: Findings from a nationwide survey in BangladeshPLOS ONE

Dear Dr. Koly,

Thank you for submitting your manuscript to PLOS ONE. After careful consideration, we feel that it has merit but does not fully meet PLOS ONE’s publication criteria as it currently stands. Therefore, we invite you to submit a revised version of the manuscript that addresses the points raised during the review process.

We look forward to receiving your revised manuscript.

Kind regards,

Hsin-Yen Yen

Academic Editor

PLOS ONE

Journal Requirements:

3. You indicated that you had ethical approval for your study. In your Methods section, please ensure you have also stated whether you obtained consent from parents or guardians of the minors included in the study or whether the research ethics committee or IRB specifically waived the need for their consent.

4. We note that Figures 3 &4  in your submission contain [map/satellite] images which may be copyrighted. All PLOS content is published under the Creative Commons Attribution License (CC BY 4.0), which means that the manuscript, images, and Supporting Information files will be freely available online, and any third party is permitted to access, download, copy, distribute, and use these materials in any way, even commercially, with proper attribution. For these reasons, we cannot publish previously copyrighted maps or satellite images created using proprietary data, such as Google software (Google Maps, Street View, and Earth). For more information, see our copyright guidelines: http://journals.plos.org/plosone/s/licenses-and-copyright.

 a. You may seek permission from the original copyright holder of Figures 3 &4 to publish the content specifically under the CC BY 4.0 license. 

Reviewers' comments:

Reviewer's Responses to Questions

**Comments to the Author**

1. Is the manuscript technically sound, and do the data support the conclusions?

Reviewer #1: Partly

Reviewer #2: Yes

2. Has the statistical analysis been performed appropriately and rigorously? 

Reviewer #1: Yes

Reviewer #2: Yes

3. Have the authors made all data underlying the findings in their manuscript fully available?

Reviewer #1: No

Reviewer #2: Yes

4. Is the manuscript presented in an intelligible fashion and written in standard English?

Reviewer #1: No

Reviewer #2: Yes

5. Review Comments to the Author

Reviewer #1: Background

1. I am a bit unsure about social change. Can you please explain or term it differently?

2. I think it would be better if the author could describe the pathways of developing mental health problems in adolescence for pre-covid period and compare it with the situation during COVID-19.

3. Study rationale could be more specifically represented in the study.

4. English editing is required

Methods

1. It would be better if the author describe a bit more about eh A2i methodology, that could give the reader a clear sense of it. Although, author have already provided some information about it but its methodology seemed not very clear to me as well as it survey design.

2. It will be helpful for the readers to get a sense of why the author had used PHQ-9 and GAD-7 scale in this survey if they describe it briefly.

3. How the author has adjusted the potential cofounder and reduce the collinearity of the regression model. Also, what have the author done to adjust the missing values, if there was any?

4. Did the author perform any model validity test?

5. A brief of categorisation process of explanatory variable would be more helpful as well as measurement process including time duration.

6. Author’s description on outcomes was very clear and specific.

7. In adolesces, behaviours changes with age, so that, I would prefer that it would be better if the author perform age-adjusted prevalence to get the more accurate results.

Discussion

It would be better if the author improves the discussion part further, discuss the findings rather repeating the results again. It required substantial revisions and English editing as well to make the findings more understandable

Reviewer #2: The manuscript was written in standard English. It used appropriate statistical technique considering the dataset leading to a technically sound finding. The conclusions and recommendations made were validly based on the findings. Yes, the data was fully available.

6. PLOS authors have the option to publish the peer review history of their article (what does this mean?). If published, this will include your full peer review and any attached files.

Reviewer #1: **Yes: **Mostaured Khan

Reviewer #2: No

---

## [Author Response · Author response to Decision Letter 0]

20 Dec 2022

Dear Dr. Hsin-Yen Yen,

We would like to thank you for your consideration of our manuscript ‘Psychosocial health of school-going adolescents during the COVID-19 pandemic: Findings from a nationwide survey in Bangladesh’ and for the feedback provided by the reviewers. We have now addressed each comment provided by the reviewers, and we think the quality of the paper has improved based on the suggestions.

Responses to Reviewer 1

Background

Reviewer’s comment: 1. I am a bit unsure about social change. Can you please explain or term it differently?

Authors’ response: We have now revised the sentence in the introduction section as follows:

“Adolescence is an important developmental transition period from childhood to adulthood that includes multiple physical, cognitive and psychosocial changes (1)”

Please check page no: 5.

Reviewer’s comment: 2. I think it would be better if the author could describe the pathways of developing mental health problems in adolescence for pre-covid period and compare it with the situation during COVID-19.

Authors’ response: Thanks for your suggestion. We have revised the introduction. 

“Depression and anxiety are the most common mental health problems in young and adolescents (2, 3).”

However, in contrast to the pre-COVID-19 era, additional stressors such as academic delays, the uncertainty of the future, financial crisis, and worry about getting infected were reported for developing mental health issues in adolescents. 

Please check page no: 5

Reviewer’s comment: 3. Study rationale could be more specifically represented in the study.

Authors’ response: Thanks for your comment. But this comment did not guide us in any direction. However, we have revised the justification of the study.

“Several studies conducted with different cohorts including general population, university students, medical students, slum-dwellers, health workers, and COVID-19 survivors highlighted various pandemic related mental health problems (e.g., anxiety, depression, stress, suicidal ideation, and behavioral problems such as problematic use of smartphone,) in Bangladesh (4-15), but research on adolescents during the COVID-19 was very limited at the time of the study.”

Please check page no: 6

Reviewer’s comment: 4. English editing is required

Authors’ response: We have two native English speakers in the authors’ list. They have now edited our manuscript. 

Methods

Reviewer’s comment: 1. It would be better if the author describe a bit more about eh A2i methodology, that could give the reader a clear sense of it. Although, author have already provided some information about it but its methodology seemed not very clear to me as well as it survey design.

Authors’ response: We have already added the following paragraph. We have also included a2i’s website (https://a2i.gov.bd/) as anyone can browse it to learn about their activity in detail. 

“a2i is a governmental program in the Information and Communication Technology (ICT) Division of Bangladesh that is supported by the Cabinet Division and United Nations Development Programme (UNDP). a2i is a program of the government's Digital Bangladesh agenda which has an edutainment online platform named "Kishore Batayan – Konnect" that has been developed for school-going adolescents [42]. a2i aims to nurture the educational, psychosocial and life skills of school-going adolescents in Bangladesh. Adolescents can share and learn essential life lessons from different creative multimedia content that can help advance their social and personal skills”.

Please check page no: 7

Reviewer’s comment: 2. It will be helpful for the readers to get a sense of why the author had used PHQ-9 and GAD-7 scale in this survey if they describe it briefly.

Authors’ response: Thanks for your suggestions. We have now added its explanation. 

“Participants’ depression and anxiety were assessed using Patient Health Questionnaire (PHQ-9) and Generalized Anxiety Disorder (GAD-7) measures as these scales were used in several studies in the Bangladesh context including school-going adolescents (16-19) .”

These scales are also affirmed to be effective and potent screening instruments for anxiety and depression in any demographic as per the previous studies (16-19). Please check page no: 9

Reviewer’s comment: 3. How the author has adjusted the potential cofounder and reduce the collinearity of the regression model. Also, what have the author done to adjust the missing values, if there was any?

Authors’ response: Before performing regression analysis, the potential multi-collinearity was checked by using tolerance (> 0.1) and variance inflation factor (VIF < 10). We had no missing values in the final analysis as we removed all missing data before the final analysis. 

Please check page no: 11

Reviewer’s comment: 4. Did the author perform any model validity test?

Authors’ response: We performed the model validity test. We have now included the values of Cox & Snell R Square, and Nagelkerke R Square for each model. 

“The values of Cox & Snell R Square, and Nagelkerke R Square for multivariable logistic regression of depression were 0.27, and 0.36, respectively.”

“The values of Cox & Snell R Square, and Nagelkerke R Square for multivariable logistic regression models of depression and anxiety were 0.21, and 0.33, respectively.”

“The values of Cox & Snell R Square, and Nagelkerke R Square for multivariable logistic regression models of depression and anxiety were 0.21, and 0.33, respectively”.

Please check page no:16-18

Reviewer’s comment: 5. A brief of categorisation process of explanatory variable would be more helpful as well as measurement process including time duration.

Authors’ response: We have now added the categorization of explanatory variables in few sections considering the word limit.

Please check page no: 10

Reviewer’s comment: 6. Author’s description on outcomes was very clear and specific.

Authors’ response: Thanks for your positive comments. 

Reviewer’s comment: 7. In adolesces, behaviours changes with age, so that, I would prefer that it would be better if the author perform age-adjusted prevalence to get the more accurate results.

Authors’ response: Thanks for your comment. We already adjusted age in each of the multivariable regression analysis. We have added the Figure 3 where the distribution of depression and anxiety with participants’ age is presented. 

Please check page no: 16

Discussion

Reviewer’s comment: It would be better if the author improves the discussion part further, discuss the findings rather repeating the results again. It required substantial revisions and English editing as well to make the findings more understandable

Authors’ response: Thanks for your suggestion. We have already added both pre-COVID-19 and during COVID-19 studies, and also compared our present findings with the previous studies. We first mentioned the present study findings briefly and then compared our findings with previous studies. However, we have now added some lines to make the discussion more comprehensive as you suggested. We have also edited the English language. 

Please check pages no: 21-23, 26-28.

Responses to Reviewer 2

Reviewer’s comment: The manuscript was written in standard English. It used appropriate statistical technique considering the dataset leading to a technically sound finding. The conclusions and recommendations made were validly based on the findings. Yes, the data was fully available.

Authors’ response: Thanks for your positive comments. We appreciate your time and review.

---

## [Decision Letter · Decision Letter 1]

8 Mar 2023

Psychosocial health of school-going adolescents during the COVID-19 pandemic: Findings from a nationwide survey in Bangladesh

PONE-D-22-07306R1

Dear Dr. Koly,

We’re pleased to inform you that your manuscript has been judged scientifically suitable for publication and will be formally accepted for publication once it meets all outstanding technical requirements.

Kind regards,

Hsin-Yen Yen

Academic Editor

PLOS ONE

Additional Editor Comments (optional):

Reviewers' comments:

Reviewer's Responses to Questions

**Comments to the Author**

1. If the authors have adequately addressed your comments raised in a previous round of review and you feel that this manuscript is now acceptable for publication, you may indicate that here to bypass the “Comments to the Author” section, enter your conflict of interest statement in the “Confidential to Editor” section, and submit your "Accept" recommendation.

Reviewer #1: All comments have been addressed

2. Is the manuscript technically sound, and do the data support the conclusions?

Reviewer #1: Yes

3. Has the statistical analysis been performed appropriately and rigorously? 

Reviewer #1: Yes

4. Have the authors made all data underlying the findings in their manuscript fully available?

Reviewer #1: Yes

5. Is the manuscript presented in an intelligible fashion and written in standard English?

Reviewer #1: Yes

6. Review Comments to the Author

Reviewer #1: I do highly appreciate the way the authors addressed the comments. I believe that the manuscript will add values to the existing literature in this field, and that readers and policymakers will have a clear understanding of the pandemic stressors on school-aged adolescents.

I have have two very minor points -

1. By suggesting the model validation test, I mainly meant to check the alpha-value, that often used to check the validity of the used scale or model against the study population characteristics. The author mentioned about Cox & Snell R Square, and Nagelkerke R Square values, which is also okay. So, I appreciate the improvements the authors made here.

2. Previously, I suggested the authors to run "age adjusted or age-standardized" prevalence analysis. The authors adjusted that regression model with age variable, which is also okay. So, I appreciate the improvements the authors made here.

Finally, thank to the authors again for addressing all the comments.

7. PLOS authors have the option to publish the peer review history of their article (what does this mean?). If published, this will include your full peer review and any attached files.

Reviewer #1: No

---

## [Editor Report · Acceptance letter]

14 Mar 2023

PONE-D-22-07306R1 

Psychosocial health of school-going adolescents during the COVID-19 pandemic: Findings from a nationwide survey in Bangladesh 

Dear Dr. Koly:

I'm pleased to inform you that your manuscript has been deemed suitable for publication in PLOS ONE. Congratulations! Your manuscript is now with our production department. 

Kind regards, 

on behalf of

Dr. Hsin-Yen Yen 

Academic Editor

PLOS ONE